# M³Seg: A Maximum-Minimum Mutual Information Paradigm for Unsupervised Topic Segmentation in ASR Transcripts

**Ke Wang[†]   Xiutian Zhao[†]   Yanghui Li   Wei Peng[*]**
Huawei IT Innovation and Research Center
{wangke215, zhaoxiutian, liyanghui, peng.wei1}@huawei.com

## Abstract

Topic segmentation aims to detect topic boundaries and split automatic speech recognition transcriptions (e.g., meeting transcripts) into segments that are bounded by thematic meanings. In this work, we propose M³Seg, a novel Maximum-Minimum Mutual information paradigm for linear topic segmentation without using any parallel data. Specifically, by employing sentence representations provided by pre-trained language models, M³Seg first learns a region-based segment encoder based on the maximization of mutual information between the global segment representation and the local contextual sentence representation. Secondly, an edge-based boundary detection module aims to segment the whole by topics based on minimizing the mutual information between different segments. Experiment results on two public datasets demonstrate the effectiveness of M³Seg, which outperform the state-of-the-art methods by a significant (18%—37% improvement) margin.

## 1 Introduction

Automatic speech recognition (ASR, also known as computer speech recognition or speech-to-text) (Rabiner and Juang, 1993; Graves and Jaitly, 2014) has brought us great convenience by transcribing conversations into text anytime and anywhere to aid human understanding. However, the generated unstructured transcriptions are sometimes too lengthy for users to grasp a high-level meaning quickly. Meeting transcripts are usually long and contain multiple heterogeneous topics in various structures, such as opening sessions, different discussion subjects, and closing sections. As one or more topics usually drive conversations or discussions, topic segmentation (Labadié and Prince, 2008) can improve the readability of transcription and facilitate

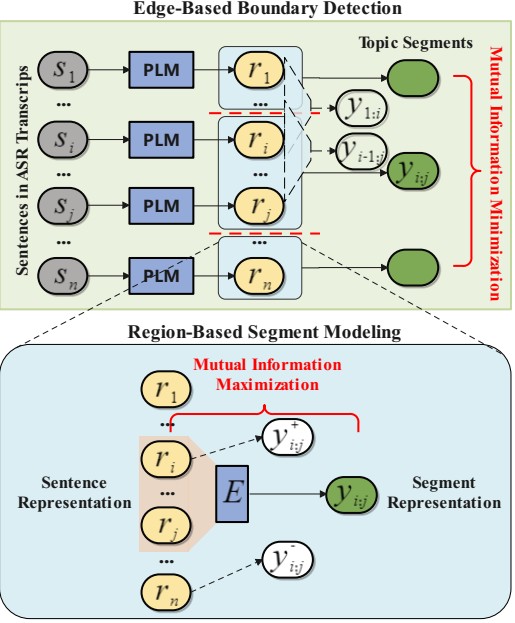

Figure 1: Illustration of M³Seg, which first learns segment representations by mutual information maximization, and then segments the entire ASR transcript by mutual information minimization. PLM refers to pre-trained language models, and dotted red lines indicate topic changes.

downstream long-text tasks such as meeting summarization, passage retrieval, and automated article generation (Mohri et al.; Feng et al.).

However, acquiring annotated training data to support topic segmentation is expensive. Unsupervised methods have attracted attention due to their less reliance on annotated data. Most existing unsupervised works can be categorized into edge-based and region-based methods. Particularly, **edge-based methods** aim to detect discontinuities (i.e., topic changes) in a sequence where semantic features change rapidly, based on word frequency (Hearst, 1997), embeddings of pre-trained language models (Solbiati et al., 2021), etc (Choi, 2000). **Region-based methods** aggregate neighbouring sentences through the homogeneity criterion, such

---

[†]Equal Contribution.
[*]Corresponding author.

as latent dirichlet allocation (Riedl and Biemann, 2012) or perplexity calculated by PLM (Feng et al.). Nevertheless, these methods rely on capturing local sentence information rather than global topic information to perform segmentation, which may be noise sensitive and have a gap in generating ground-truth semantic boundaries.

In contrast to prior works, which fail to capture the global semantic-level topic information of different segments, we propose a novel **Maximum-Minimum Mutual** information paradigm for unsupervised topic **seg**mentation ($\text{M}^3\text{Seg}$). Our work is inspired by mutual information, which measures how much one random variable tells us about another. Intuitively, sentences in the same segment should depend on the same topic (with high mutual information), and different topic segments should be independent of each other (with low mutual information). Based on this insight, $\text{M}^3\text{Seg}$ divides an ASR script into different topic segments based on a two-stage process: region-based segment modeling and edge-based boundary detection. We first learn a segment encoder by maximizing the mutual information between the global segment representation and each local contextualized sentence representation (provided by pre-trained language Models). Stage 2 detects topic boundaries by minimizing mutual information between different segments.

Experimental results on two widely-used benchmark datasets show that $\text{M}^3\text{Seg}$ consistently surpasses five existing methods by a wide margin. We conduct ablation studies to demonstrate the effectiveness of the proposed model and show the great utility of using mutual information in this task.

## 2 Method

Formally, let $s$ denote a meeting transcript produced by an automatic speech recognition (ASR) system, which consists of a list of $n$ utterances $s = \{s_1, s_2, \cdots, s_n\}$. Topic segmentation can be seen as a problem of topic change detection, and aims to cut the transcript into consecutive segments $\{s_{1:i-1}, s_{i:j}, \cdots, s_{k:n}\}$ based on the underlying topic structure. For each segment $s_{i:j}$, it represents a segment in a meeting transcript, from the $i$-th sequence to the $j$-th sequence $s_{i:j} = \{s_i, \cdots, s_j\}$.

For example, an hour-long meeting transcript can be broken down into different topic segments (e.g., opening sessions, different discussion subjects, and closing remarks) to make it more readable. Note that we cannot access gold-standard segments as human annotations do not exist.

**Model Overview**   As aforementioned, given a transcript $s = \{s_1, s_2, \cdots, s_n\}$, topic segmentation seeks to splits it into several topic segments, where the sentences in each segment belong to the same topic (*requirement i*) but different segments represents relatively independent topics (*requirement ii*). Inspired by this idea, we propose a maximum-minimum mutual information paradigm for topic segmentation (titled $\text{M}^3\text{Seg}$) in an unsupervised manner, which learns segment representations by maximizing mutual information and partitions different segments by minimizing mutual information.

Mutual information (MI) can measure the dependence between two random variables (Shannon, 1948). Given two random variables $a$ and $b$, the MI between them is $\mathcal{I}(a;b) = \sum_{a,b} P(a,b) \log \frac{P(a|b)}{P(a)}$. The intuitive interpretation of $\mathcal{I}(a;b)$ is a measurement of the degree $a$ reduces the uncertainty in $b$ or vice versa. For example, the MI $I(a;b)$ is equal to 0 when $a$ and $b$ are independent. Therefore, from the view of MI, different topic segments should be as independent of each other as possible (i.e., MI minimization), but sentences within the same topic segment should be as dependent on the same topic as possible (i.e., MI maximization). Based on this insight, $\text{M}^3\text{Seg}$ consists of two stages (as depicted in Figure 1):

**(1) Segment Modeling Based on MI Maximization**   Intuitively, knowing a sentence reduces the uncertainty of its corresponding topic. Thus, we train a segment encoder $E$ to learn the global segment representation $y_{i:j} = E_\theta(r_{i:j})$ of the segment $s_{i:j}$ by maximizing the MI between it and each of its local sentence representations $r_k, k \in [i,j]$ (*requirement i*):

$$\mathcal{J}_{SM} = \max \sum_{i,j,k \in [1,n], k \in [i,j]} \mathcal{I}(r_k; y_{i:j}) \quad (1)$$

Specifically, given $n$ utterances $s = \{s_1, s_2, \cdots, s_n\}$ in an ASR transcript, we first use the pre-trained language model (**PLM**) to obtain the contextualized representation $r$ of each sentence (Peters et al., 2018): $r = \{r_1, \cdots, r_i, \cdots, r_n\}, r_i = PLM(s_i), s_i \in \mathbb{R}^d$, where $d$ is the representation dimension of PLM. In our case, $r_i$ is computed by applying a mean-over-time pooling layer on the token

representations of the last layer of RoBERTa-base ([Liu et al., 2019](#)) model. We then use the segment encoder $E$ to get the segment representation $y_{i:j}$ of the text segment $s_{i:j}$: $y_{i:j} = E_\theta(r_{i:j})$, $y_{i:j} \in \mathbb{R}^d$, where $\theta$ denotes the parameters of $E$. The segment encoder $E$ is trained by maximizing the MI between the global segment representation and each of its local sentence representations. However, mutual information estimation is generally intractable for continuous and high dimensional random variables, so we maximize the InfoNCE ([Logeswaran and Lee, 2018](#)) lower bound estimator of Eq 1. Following ([Kong et al., 2020](#)), maximizing InfoNCE is analogous to maximizing the standard cross-entropy loss: $\mathcal{L}_{sm}\& = -\mathbb{E}_{i,j\in[1,n],i\le j}[\log \frac{\exp(y_{i:j} \circ y_{i:j}^+)}{\exp(y_{i:j} \circ y_{i:j}^+) + \exp(y_{i:j} \circ y_{i:j}^-)}]$, where $y_{i:j}^+ \in r_{i:j}$ is sampled from sentence representations in the segment $s_{i:j}$, and $y_{i:j}^-$ is sampled from the rest. We use the dot product between embeddings to measure the distance (i.e., $\circ$) in the vector space. Note that we only need to train the segment encoder $E$, while PLM's parameters are fixed.

Our assumption is that in a meeting transcript, a continuous sequence of utterances contains a thematic information, which can be either coarse-grained or fine-grained. For example, the entire meeting transcript's utterances may be based on a certain motivation (coarse-grained theme) for convening, while if divided into fine-grained segments, each utterance can be associated with a certain issue (fine-grained theme). It is based on this assumption that we construct data from any continuous sequence of utterances in the same meeting transcript and use the maximization of mutual information to train a segment encoder $E$ to learn the global segment representation (Eq 1) ([Wang and Wan, 2020, 2021](#)). This also contributes to the effectiveness of our method.

**(2) Boundary Detection Based on MI Minimization** After obtaining the segment representation of any region, we propose an edge-based boundary detection module to detect topic changes by minimizing the MI between the segment representations of different topic segments (*requirement ii*): $\mathcal{J}_{BD} = \min \sum_{i,j,k\in[1,n],i\ne k} \mathcal{I}(y_{i:j}; y_{j:k})$. Specifically, as we are primarily interested in maximizing the MI gap, and not concerned with its precise value, we can rely on non-Kullback–Leibler diver-

gences which may offer favorable trade-offs.

Following the $\text{Info}_{NCE}$ estimator in [Poole et al. (2019)](#), we define a Jensen-Shannon mutual information estimator of two factorized latent variables $a$ and $b$: $\hat{\mathcal{I}}^{JSD}(a; b)\& := \sum_{a,b}[\log(1 + exp(a \circ b))] - \sum_{a,b^-}[\log(1 + exp(a \circ b^-))]$, where $b^-$ is sampled from the complement set $\bar{b}$ of $b$. In order to achieve a single-pass computation (low time complexity), we calculate the difference in MI between the proposed boundary $i$ and its adjacent offsets of +/- 1 as the MI bound.

$$MIG(y_{1:i-1}; y_{i:j}) := \underbrace{y_{1:i-2} \circ y_{i-1:j} - y_{1:i-1} \circ y_{i:j}}_{Disentanglement\ with\ offset-1}$$
$$+ \underbrace{y_{1:i} \circ y_{i+1:j} - y_{1:i-1} \circ y_{i:j}}_{Disentanglement\ with\ offset+1} . \quad (2)$$

This can be easily extended to more complex bounds in the future. As a consequence, we propose a metric based on MI gap (MIG) to quantitatively assess the effectiveness of disentanglement between two neighboring regions $s_{1:i-1}$ and $s_{i:j}$: $MIG(y_{1:i-1}; y_{i:j}) := y_{1:i} \circ y_{i+1:j} + y_{1:i-2} \circ y_{i-1:j} - 2 * y_{1:i-1} \circ y_{i:j}$.

Finally, we derive the topic boundaries as pairs of regions $y_{1:i-1}$ and $y_{i:j}$ where $MIG(y_{1:i-1}; y_{i:j})$ scores are greater than a certain threshold $\delta$. The MIG measures how much the segment representations change when we move the boundary by one position. A positive MIG means that the segments become more dissimilar when we shift the boundary, indicating a potential topic change. A negative MIG means that the segments become more similar when we shift the boundary, indicating a coherent topic. Therefore, we can use the MIG as a criterion for detecting topic boundaries.

## 3 Experiments

To demonstrate the effectiveness of our model, we evaluate $M^3$Seg on two widely-used benchmark datasets, AMI Meeting Corpus ([Carletta et al., 2005](#)) and the ICSI Meeting Corpus (the dataset details are shown in **Appendix A**.). According to the paper [Ghosh et al. (2022)](#), it is more likely that the dataset we used consists of Unstructured Chats, transcriptions of spoken language from a spoken scene, which result in often incoherent and incomplete utterances with some word errors.

We compare $M^3$Seg with several state-of-the-art unsupervised topic segmentation methods, including region-based methods (i.e., **TopicTiling** ([Riedl and Biemann, 2012](#)) and **DialoGPT** ([Feng et al.](#)) ) and edge-based methods (i.e., **TextTiling** ([Hearst, 1997](#)), **C99** ([Choi, 2000](#)), and **TextTiling-BERT**

(Solbiati et al., 2021)). For analysis, we also show the **Random** and **Even** methods, which refer to placing topic boundaries randomly and every $n$-th utterance, respectively. We perform two standard evaluation metrics, **Pk** (Beeferman et al., 1999) and WinDiff (**Wd**) (Pevzner and Hearst, 2002) scores (details are shown in **Appendix B**), and the model implementation details are shown in **Appendix C**.

Regarding the detection of multiple boundaries, we use a certain threshold $\delta$ to control it. That is, as long as the mutual information gap between pairs of regions is greater than $\delta$, it will be considered as topic boundaries (detailed in Section 2). For the random baseline, following the settings of TextTiling-BERT, we set each utterance to have a 0.30 probability of being placed with topic boundaries (the reason is that this value is roughly consistent with the probability of boundaries occupying the entire utterance).

| Method | AMI | | ICSI | |
|---|---|---|---|---|
| | **Pk** | **Wd** | **Pk** | **Wd** |
| Random | 0.570 | 0.747 | 0.640 | 0.981 |
| Even | 0.518 | 0.559 | 0.642 | 0.924 |
| TextTiling | 0.412 | 0.426 | 0.426 | 0.600 |
| C99 | 0.442 | 0.457 | 0.514 | 0.550 |
| TopicTiling | 0.356 | 0.360 | 0.351 | 0.380 |
| DialoGPT | 0.355 | 0.356 | 0.319 | 0.333 |
| TextTiling-BERT | 0.352 | 0.352 | 0.324 | 0.343 |
| M$^3$Seg (Ours) | **0.248** | **0.286** | **0.203** | **0.224** |

Table 1: Results of topic segmentation. The lower the values of **Pk** and **Wd**, the higher the segment performance. We mark the best results in bold.

**Main Results** Table 1 compares the results of baselines and our method on two meeting transcription datasets. From the results, we can see that: 1) TopicTiling has a significant improvement over TextTiling, which shows that considering the global region-based topic information can facilitate topic segmentation. 2) DialoGPT and TextTiling-BERT achieve lower error rates than word-frequency-based approaches (i.e., TextTiling and C99), reflecting that the representations provided by PLMs provide better contextual information. 3) Nevertheless, our model outperforms all baseline models on two datasets by a wide margin (about +18% $\sim$ +37% reduction in error rate), demonstrating the effectiveness of using MI for topic segmentation.

**Module Effectiveness Analysis** To investigate the importance of the model's individual components, we perform ablations by removing the region-based segment modeling module and edge-

| Model Variants | AMI | | ICSI | |
|---|---|---|---|---|
| | **Pk** | **Wd** | **Pk** | **Wd** |
| M$^3$Seg | 0.248 | 0.286 | 0.203 | 0.224 |
| w/o segment modeling | 0.363 | 0.379 | 0.358 | 0.363 |
| w/o boundary detection | 0.309 | 0.338 | 0.285 | 0.306 |

Table 2: Ablation study of different components. "w/o" means "without".

| PLM Variants | AMI | | ICSI | |
|---|---|---|---|---|
| | **Pk** | **Wd** | **Pk** | **Wd** |
| BERT-base | 0.325 | 0.334 | 0.308 | 0.326 |
| BERT-large | 0.237 | 0.303 | **0.198** | 0.245 |
| XLNet | 0.307 | 0.319 | 0.3248 | 0.340 |
| RoBERTa | **0.248** | **0.286** | 0.203 | **0.224** |

Table 3: Results of **M$^3$Seg** with different PLMs.

based boundary detection module. For "w/o segment modeling", we directly apply a max-over-time pooling operation to sentences instead of the segment encoder $E$. For "w/o boundary detection", we apply TextTiling (Hearst, 1997) algorithm, which calculates the similarity based on the segment representations provided by $E$. From Table 2, two components play a role, yet the most significant drop when the region-based segment modeling module is removed, demonstrating the great effectiveness of using MI to model segments.

**Influence of Pre-trained Language Models (PLMs)** To investigate the importance of the PLM's contextualized representations, we perform ablations by using different PLMs in Table 3. From the results, we can see that the larger the pre-trained model leads to more significant improvement, indicating that the contextual information provided by PLM with more data and parameters is more conducive to topic segmentation. But **M$^3$Seg** can bring consistent and significant improvements under different pre-trained language models, reflecting the effectiveness of our method.

| Sentence Representation Variants | AMI | | ICSI | |
|---|---|---|---|---|
| | **Pk** | **Wd** | **Pk** | **Wd** |
| *CLS*-Embedding | 0.347 | 0.365 | 0.342 | 0.358 |
| Max-Pooling | 0.264 | 0.297 | 0.316 | 0.339 |
| Mean-Pooling | **0.248** | **0.286** | 0.203 | 0.224 |
| Sentence-BERT | 0.267 | 0.303 | **0.197** | **0.221** |

Table 4: Results of **M$^3$Seg** with different contextualized sentence representations.

**Influence of Sentence Representations ($r$)** To analyze the influence of different sentence representations of a PLM, we conduct the ablation experiments with different contextualized repre-

sentation calculation methods, including: 1) **CLS-Embedding**: The representation of the first token *[CLS]* is used as the contextualized representation of the entire sentence. 2) **Max-Pooling**: We apply a max-over-time pooling layer on the token representation of the last layer of a PLM. 3) **Mean-Pooling**: Similarly, we apply a mean-over-time pooling layer on the token representations. 4) **Sentence-BERT**: We use Sentence-BERT (Reimers and Gurevych, 2019) embedding as our contextualized sentence representation. From the results in Table 4, *CLS*-Embedding performs the worst, indicating the importance of considering global information in topic segmentation. Overall, all methods achieve competitive performances, especially Mean-Pooling and Sentence-BERT, demonstrating the effectiveness of the contextualized sentence representation provided by PLMs.

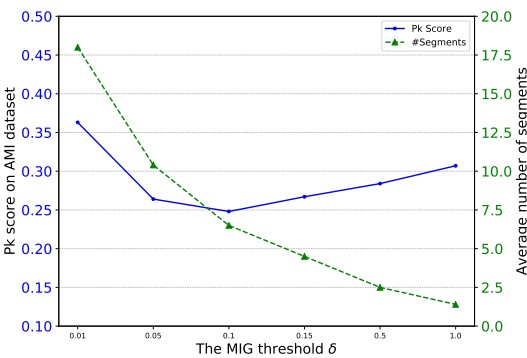

Figure 2: Results of different MIG threshold $\delta$.

**Influence of the MIG Threshold ($\delta$)** We also investigate the impact of different MI gap threshold $\delta$ on the results. Figure 2 presents the Pk score and the number of topic segments after segmentation of our model on the AMI dataset under different MIG threshold $\delta$. With the increase of $\delta$, the number of segments decreases, indicating that the larger the threshold of the MI gap leads to the rougher segmentation of topics. Interestingly, the Pk score first decreases to a minimum and then increases, potentially due to the presence of an optimal decision point wrt the selection of MIG threshold $\delta$.

## Limitations

In this study, we introduce the M³Seg framework, which is a novel approach to unsupervised topic segmentation. We transform the task into an optimization problem that maximizes intra-segment mutual information and minimizes inter-segment mutual information. Our experiments and analysis

demonstrate that the proposed model outperforms competitive systems by reducing error rates by 18% to 37%. These results emphasize the effectiveness of using mutual information for topic segmentation and suggest future opportunities to develop more complex and controllable systems.

However, our approach has limitations that should be acknowledged. Firstly, it has only been tested on the English language, and further experimentation is required to evaluate its performance on low-resource languages. Additionally, our method relies on pre-trained language models, which may not always be available or suitable for certain applications. Nevertheless, we believe that our maximum-minimum mutual information paradigm has potential to advance the development of unsupervised topic segmentation systems.

## Acknowledgement

We would like to express our gratitude to Rongjun Li, Suqing Yan, Xupeng Meng and Wei Huang from Huawei for their invaluable assistance and detailed discussions on the ideas during the implementation process. Their expertise and insights have been instrumental in the success of our project. We also wish to extend our appreciation to the anonymous reviewers whose constructive feedback greatly improved our work. Their comments and suggestions were of immense help throughout the course of our research.

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

# Appendix

## A Datasets

The AMI Meeting Corpus (Carletta et al., 2005) and the ICSI Meeting Corpus (Janin et al., 2003) include 139 and 75 transcribed meetings with topic segmentation annotations, respectively. The word error rates of ASR transcription for AMI and ICSI are 36% and 37%, respectively. We use the human-transcribed reference (gold) transcripts as same as in (Solbiati et al., 2021), which are corrected by humans based on the ASR transcript (which seem to be missing in AMI and ICSI), but still contain many errors, such as grammatical errors like "me and him have done this". We use all the data as the test set, and consider only the top-level meeting changes (i.e., linear topic segmentation). Noted that we used each script to train the segment encoder $E$, but we did not use the annotations, which conforms to the unsupervised setting.

## B  Metrics

Following previous works (Solbiati et al., 2021), we perform two standard evaluation metrics, **Pk** (Beeferman et al., 1999) and WinDiff (**Wd**) (Pevzner and Hearst, 2002) scores, to show the performance of segmented results. Both metrics use a fixed sliding window over the document, and calculate the segmentation error by comparing the number of boundaries in the ground truth with the number of boundaries predicted by the topic segmentation model.

## C  Implementation

Our segment encoder $E$ is a one-layer Transformer (Vaswani et al., 2017) with a dimension of $d = 768$. The threshold $\theta$ of the mutual information gap score is set to 0.01. The pre-trained language model uses RoBERTa-base (Liu et al., 2019) and can be easily migrated to other PLM models. We implement our model based on PyTorch and use two Tesla V100 graphic cards for learning. In order to use the original features obtained from the PLM without additional scaling, we set Dim(E) to the same dimension as the output of layer of PLM.

We train E separately for each meeting input, using only the input text list and random segment boundaries to maximize mutual information as the training goal. No gold segment boundaries are required, which is consistent with the test scenario, that is, it is not biased by other meeting data.