# OpenReview forum: "M$^3$Seg: A Maximum-Minimum Mutual Information Paradigm for Unsupervised Topic Segmentation in ASR Transcripts"
_EMNLP/2023/Conference — EMNLP 2023 Main_

### Official Review · Reviewer_bouu · 2023-08-02

**Soundness:** 4

**Excitement:**

4: Strong: This paper deepens the understanding of some phenomenon or lowers the barriers to an existing research direction.

**Paper Topic And Main Contributions:**

Main contributions of the paper are:

1. Mutual information based framework that incorporates region-based and edge-based techniques for topic segmentation of a document. This framework captures both the local and global topic level semantic information.
2. Experimental evidence of the effectiveness of the framework compared to other SoTA methods on two public meeting transcripts datasets.
3. Ablation studies to study the effect of region based and edge based techniques' contribution to the overall performance of the framework, as well as, ablation studies of different sentence representation models.
4. Clearly articulated ideas, and well-presented material.

**Questions For The Authors:**

1. During training, did you train the segment encoder over all possible segments? It seems so because you don't use the annotation data to know the start and end sentences of the segments.

**Reasons To Accept:**

1. The framework's region based and edge based approaches are well-motivated, and necessary references are provided for the reader to understand the two approaches.
2. The focus on local and global semantics, and their individual effects via ablation study of segment modeling and boundary/edge detection respectively are explained and studies clearly.
3. Since the framework can be used with any sentence representation models, an ablation study of the effect of choice of pretrained language model is done and the final choice of model is explained.

**Reasons To Reject:**

1. The segment encoder would need to encode all possible segments during training as well as inference. There are $n(n-1)$ number of segments. The quadratic effect is not explained, and it's not clear how it affects time to train.
2. The framework was only applied to meeting transcripts, and the title aptly states that. However, it is not clear what property of meeting transcripts differentiate them from long form documents. Given this, it indicates the lack of generalizability of this approach.

**Reproducibility:**

4: Could mostly reproduce the results, but there may be some variation because of sample variance or minor variations in their interpretation of the protocol or method.

**Reviewer Confidence:**

3: Pretty sure, but there's a chance I missed something. Although I have a good feel for this area in general, I did not carefully check the paper's details, e.g., the math, experimental design, or novelty.

---

> ### Author Rebuttal · Authors · 2023-08-29
>
> Thank you for your insightful comments and suggestions. We are glad that you found our framework well-motivated and our focus on local and global semantics clearly explained.
>
> ***
>
> > During training, did you train the segment encoder over all possible segments? It seems so because you don't use the annotation data to know the start and end sentences of the segments. The segment encoder would need to encode all possible segments during training as well as inference. There are number of segments. The quadratic effect is not explained, and it's not clear how it affects time to train.
>
>
> Yes, you are correct. In our work, we adopted a simple quadratic construction of all possible segments to train the segment encoder for validating the effectiveness of our M3Seg topic segmentation without the annotation data.
>
> We randomly sampled different proportions of data from the all possible segments data for training the segment encoder during inference, and the results are shown in the table below. We also provide the average time cost per case for analysis:
>
> | Scale of the Dataset | AMI-Pk | AMI-Wd | ICSI-Pk | ICSI-Wd | Time Cost/Case |
> |:--------------------:|:------:|:------:|:-------:|:-------:|:--------------:|
> |          0%          |  0.332 |  0.329 |  0.319  |  0.331  |      11ms      |
> |          25%         |  0.277 (+16.7%) |  0.297(+9.7%) |  0.262(+17.9%)  |  0.289(+12.7%)  |      49ms      |
> |          50%         |  0.259 (+6.4%)|  0.291(+2.0%) |  0.232(+11.5%)  |  0.247(+14.5%)  |      70ms      |
> |         100%         |  0.248 (+4.2%)|  0.286(+1.7%) |  0.203(+12.5%)  |  0.224(+9.3%)   |     124 ms     |
>
> The first row of the table, with a scale of the dataset at `0%`, is a special cross-domain example, where we trained the segment encoder on ICSI data and tested it on AMI data, and vice versa. From the results, we can see that from `0%` to `25%`, the model performance increased significantly (average increase of `16.7%,9.7%,17.9%,12.7%`), indicating that using the inference samples themselves for training is very helpful.
>
> Secondly, as the proportion of data increases from `50%` to `100%`, the increase in performance is relatively low, indicating that efficient data selection strategies for training the segment encoder are worth further research. From the results in the table, we can see that time cost is proportional to both training data and performance, so this is a trade-off strategy that depends on specific scenarios.
>
> ***
>
> > The framework was only applied to meeting transcripts, and the title aptly states that. However, it is not clear what property of meeting transcripts differentiate them from long form documents. Given this, it indicates the lack of generalizability of this approach.
>
> Thank you for your suggestion. We have tested the performance of our framework on other types of documents, such as structured, semi-structured, and unstructured conversational text. Following the setting of the paper [1], we conducted experiments on the Topical Chat, BOLT, and Wiki-727K datasets proposed in the paper, and the experimental results are as follows:
>
> | Method  | Topical Chat |    BOLT   | Wiki-727K |
> |----------------------------|:------------:|:---------:|:---------:|
> | Cross Segment BERT|     0.764    |   0.569   |   0.604   |
> | Cross Segment RoBERTa-Base|     0.767    |   0.561   |   0.599   |
> | Hierarchical Bi-LSTM|   **0.951**  |   0.443   |   0.570   |
> | M3Seg|     0.950    | **0.602** | **0.612** |
>
> As can be seen from the results, our method has good generalizability and achieves performance comparable to these supervised methods that use domain data for fine-tuning, even though our method is unsupervised. We will add this analysis to our paper in our revised version.
>
> [1] Ghosh et al. 2022. Topic Segmentation in the Wild: Towards Segmentation of Semi-structured & Unstructured Chats.
> ***
>
> Thank you for all the great questions and suggestions! Please let us know if you have any remaining concerns.

---

### Official Review · Reviewer_t5HK · 2023-08-03

**Soundness:** 4

**Excitement:**

3: Ambivalent: It has merits (e.g., it reports state-of-the-art results, the idea is nice), but there are key weaknesses (e.g., it describes incremental work), and it can significantly benefit from another round of revision. However, I won't object to accepting it if my co-reviewers champion it.

**Paper Topic And Main Contributions:**

The paper introduces a new algorithm to perform unsupervised segmentation of sentences into different topics. With experiments on two datasets and comparisons with previous methods, the paper shows that the proposed methodology provides significant improvements.

**Questions For The Authors:**

How do you define the i:j segments in paragraph 1) of section 2?

**Reasons To Accept:**

The results seems to demonstrate the effectiveness of the approach, and the reasons of the improvements are analyzed in details, including the robustness to different PLMs.

**Reasons To Reject:**

Confidence intervals are missing, and the paper is not easy to read for someone who is not expert of the topic.

**Reproducibility:**

3: Could reproduce the results with some difficulty. The settings of parameters are underspecified or subjectively determined; the training/evaluation data are not widely available.

**Reviewer Confidence:**

1: Not my area, or paper was hard for me to understand. My evaluation is just an educated guess.

**Typos Grammar Style And Presentation Improvements:**

line 51: etc might be explained instead.
line 110: representing -> represents.
Some references are missing the year, so bibliography should be more careful checked.

---

> ### Author Rebuttal · Authors · 2023-08-29
>
> Thank you for your valuable and constructive comments on our paper. We appreciate your recognition of the effectiveness of our approach and the detailed analysis of the reasons for the improvements, including the robustness to different PLMs. Below are our responses to your specific comments.
>
> ***
>
> > Confidence intervals are missing, and the paper is not easy to read for someone who is not expert of the topic.
>
> We thank the reviewer for pointing out this issue. Regarding the missing confidence intervals, we would like to clarify that our mehtod is inspired by the mutual information theory yet does not apply a predetermined confidence interval when sampling. However, we will incorpate this in our future work and suppliement the experiemnt by testing the method under different confidence boundary settings.
>
> We have reorganized the structure of our paper and include more introductory contents to ensure a more smooth reading experience in our revised version.
>
> ***
>
> > How do you define the i:j segments in paragraph 1) of section 2?
>
> We would like to clarify that the notation S_i:j represents a segment in a meeting transcript, from the i-th sequence to the j-th sequence. We apologize for any confusion and will add more explanation about S_i:j to our paper. We have also carefully revise our paper, including checking the bibliography, to make it more understandable.
>
> ***
>
> We hope that these clarifications address your questions. Please let us know if you have any remaining concerns.

---

### Official Review · Reviewer_ZX6U · 2023-08-12

**Soundness:** 3

**Excitement:**

3: Ambivalent: It has merits (e.g., it reports state-of-the-art results, the idea is nice), but there are key weaknesses (e.g., it describes incremental work), and it can significantly benefit from another round of revision. However, I won't object to accepting it if my co-reviewers champion it.

**Paper Topic And Main Contributions:**

This work proposes an approach to tackle the topic segmentation problem, which splits a long spoken transcript into several regions with similar topics, the proposed approach consists of two steps taking advantage of mutual information, in the first step, it learns segment encoder which encourages embedding of sentences within the same region to be similar, in the 2nd step, it uses those embedding to segment transcripts based on the "mutual information gap". The proposed approach outperforms a few benchmarks significantly.

**Reasons To Accept:**

The main motivation is interesting, which tries to maximize the mutual information within the same region while minimizing the mutual information across regions.

**Reasons To Reject:**

the proposed methods of both maximization and minimization are ambiguous and not very well aligned with the original mutual information motivation. For example, In maximization formula 1),  i,j are iterated from 1 to n, which implies that they are just random boundaries and not the actual region boundary, then maximizing mutual information with this random region might not be very helpful. In the minimization formula 2, it is simply a gap between representations, it is difficult to claim they are based on mutual information as there are many skips from requirement 2.

Many details are not covered in the main section. For example, the segment encoder is one of the most important ideas in this work, but its implementation is totally omitted and only mentioned in the appendix. It is also not clear how to detect multiple boundaries in the text for the proposed approach. Details about other baselines are also not enough, For example, for the random baseline, what's the probability to place the boundary?

**Reproducibility:**

3: Could reproduce the results with some difficulty. The settings of parameters are underspecified or subjectively determined; the training/evaluation data are not widely available.

**Reviewer Confidence:**

3: Pretty sure, but there's a chance I missed something. Although I have a good feel for this area in general, I did not carefully check the paper's details, e.g., the math, experimental design, or novelty.

---

> ### Author Rebuttal · Authors · 2023-08-29
>
> Thank you for your comments and feedback on our paper. We appreciate your recognition of our motivation to maximize the mutual information within the same region while minimizing the mutual information across regions, and we are encouraged by your finding it interesting. We would like to address some potential misunderstandings in your comments.
>
> ***
>
> > In maximization formula 1), i,j are iterated from 1 to n, which implies that they are just random boundaries and not the actual region boundary, then maximizing mutual information with this random region might not be very helpful.
>
> We would like to clarify our hypothesis. Our assumption is that in a meeting transcript, a continuous sequence of utterances contains a thematic information, which can be either coarse-grained or fine-grained. For example, the entire meeting transcript’s utterances may be based on a certain motivation (coarse-grained theme) for convening, while if divided into fine-grained segments, each utterance can be associated with a certain issue (fine-grained theme). It is based on this assumption that we construct data from any continuous sequence of utterances in the same meeting transcript and use the maximization of mutual information to train a segment encoder E to learn the global segment representation (Formula 1). This also contributes to the effectiveness of our method.
>
> ***
>
> > In the minimization formula 2, it is simply a gap between representations, it is difficult to claim they are based on mutual information as there are many skips from requirement 2.
>
> We would like to clarify that Formula 2 describes a simple method for determining topic segmentation boundaries based on maximizing the mutual information gap between adjacent segments, given that we have trained a good segment encoder E that can capture the thematic annotations in any continuous sequence of sentences (i.e., Formula 1). As an initial study, we did indeed propose a simple method to analyze the effectiveness of our entire mutual information learning framework. This can be easily extended to more complex bounds in the future.
>
> ***
>
> > The proposed methods of both maximization and minimization are ambiguous and not very well aligned with the original mutual information motivation.
>
> We were inspired by the idea that mutual information (MI) can measure the dependence between two random variables, and proposed a general framework for unsupervised topic segmentation using the inherent characteristics of meeting transcript data. However，we would like to clarify that as an initial study on unsupervised topic boundary segmentation using mutual information, we used relatively simple settings for each part of our M3Seg framework in order to analyze its effectiveness (e.g., constructing random continuous sentence regions to train the topic encoder and using the method of maximum adjacent mutual information difference to determine topic boundaries). As you rightly pointed out, it is indeed possible to explore more complex data construction and mutual information gap-based boundary detection methods. However, this precisely demonstrates the robustness of our method and leaves room for improvement in future work.
>
> ***
>
> > Many details are not covered in the main section. The segment encoder is one of the most important ideas in this work, but its implementation is totally omitted and only mentioned in the appendix. It is also not clear how to detect multiple boundaries in the text for the proposed approach. Details about other baselines are also not enough, For example, for the random baseline, what's the probability to place the boundary?
>
> As you mentioned, while we introduced the core training method of the segment encoder (Equation 1) in the paper, we placed the specific implementation details of the segment encoder in the appendix.
>
> Regarding the detection of multiple boundaries, we use a certain threshold δ to control it. That is, as long as the mutual information gap between pairs of regions is greater than δ, it will be considered as topic boundaries (detailed in Section 2). For the random baseline, following the settings of TextTiling-BERT, we set each utterance to have a 0.30 probability of being placed with topic boundaries (the reason is that this value is roughly consistent with the probability of boundaries occupying the entire utterance).
>
> We have rewritten our paper and placed more core information in the main section in line with your comments. We will certainly leverage the one more page in the final version to provide more details, elease all code and data, including our methods and the datasets used for reproducing the results, should we been given the opportunity.
>
> ***
>
> Thank you again for your valuable feedback! Hopefully this clarification addresses your questions. Please let us know if you have any remaining concerns,

---

### Official Review · Reviewer_mmRA · 2023-08-12

**Soundness:** 4

**Excitement:**

4: Strong: This paper deepens the understanding of some phenomenon or lowers the barriers to an existing research direction.

**Missing References:**

Authors, please cite https://arxiv.org/abs/2211.14954 - the paper shows work on topic segmentation methods for unstructured and semi-structured conversations.

**Paper Topic And Main Contributions:**

The authors in this paper present a novel approach to unsupervised topic segmentation called M^3Seg. The paper addresses the problem of detecting topic boundaries and splitting automatic speech recognition (ASR) transcripts into meaningful segments. The main contribution of the authors of this paper is the development of a maximum-minimum mutual information paradigm for unsupervised topic segmentation. This approach transforms the task into an optimization problem that maximizes intra-segment mutual information and minimizes inter-segment mutual information. Experimental results demonstrate that the proposed model outperforms competitive systems by reducing error rates by 18% to 37%.

**Questions For The Authors:**

Would M3Seg work with transcriptions where utterances are incoherent and incomplete (no fully formed sentences)?

**Reasons To Accept:**

This paper introduces a novel approach to unsupervised topic segmentation, called M3Seg, which is based on a maximum-minimum mutual information paradigm. The paper's results demonstrate that the proposed approach outperforms competitive systems by reducing error rates by 18% to 37%. These results emphasize the effectiveness of using mutual information for topic segmentation and suggest future opportunities to develop more complex and controllable systems. This paper would benefit the NLP/ASR community by providing a new and effective approach to unsupervised topic segmentation.

**Reasons To Reject:**

Comparing results from Table 2 and 3, the Pk and the WinDiff values are marginally but not notably different for M3Seg. Along with the identified limitation of expanding beyond English transcriptions, especially to low-resource languages, authors should also look into defining whether the meeting transcript datasets are structured conversations or unstructured conversations. https://arxiv.org/abs/2211.14954 by Ghosh et. al found that topic segmentation methods for unstructured conversations (conversations that are ill-formed, short sentences, and incoherent, which is a possibility in meeting transcripts but could be more evident in casual conversations) require advanced modeling techniques from structured conversations.

**Reproducibility:**

3: Could reproduce the results with some difficulty. The settings of parameters are underspecified or subjectively determined; the training/evaluation data are not widely available.

**Reviewer Confidence:**

5: Positive that my evaluation is correct. I read the paper very carefully and I am very familiar with related work.

**Typos Grammar Style And Presentation Improvements:**

There are some grammatical errors, such as subject-verb agreement (e.g., 'two components play a role' instead of 'two components play roles'). The paper could benefit from more consistent formatting, such as consistent use of capitalization in headings and subheadings. Some sentences could be rephrased for clarity, such as 'The threshold θ of the mutual information gap score is set to 0.01' could be rephrased as 'The mutual information gap score threshold, denoted by θ, is set to 0.01'.

---

> ### Author Rebuttal · Authors · 2023-08-28
>
> Thank you for your positive and constructive feedback! We are glad that you found our apporach new and effective. Below are our responses to specific comments.
>
> ***
>
> > Comparing results from Table 2 and 3, the Pk and the WinDiff values are marginally but not notably different for M3Seg.
>
> |                        |     AMI-Pk    |     AMI-Wd    |    ICSI-Pk    |    ICSI-Wd    |
> |------------------------|:-------------:|:-------------:|:-------------:|:-------------:|
> | M3Seg                  |     0.248     |     0.286     |     0.202     |     0.224     |
> | w/o segment modeling   |  0.363(46.4%↓) | 0.379(32.5%↓) | 0.358(77.2%↓) | 0.363(62.1%↓) |
> | w/o boundary detection | 0.309(25.0%↓) | 0.338(18.2%↓) | 0.285(41.1%↓) | 0.306(36.6%↓) |
>
> Table 2 above demonstrates an ablation study removing different components, and the small difference in the results indicates that both parts are quite effective.
>
>
> |            |    AMI-Pk    |    AMI-Wd    |    ICSI-Pk   |    ICSI-Wd   |
> |------------|:------------:|:------------:|:------------:|:------------:|
> | BERT-base  |     0.325    |     0.334    |     0.308    |     0.326    |
> | BERT-large | 0.237(27.1%↓) |  0.303(9.3%↓) | 0.198(35.7%↓) | 0.245(24.8%↓) |
> | XLNet      |  0.307(5.5%↓) |  0.319(4.5%↓) |  0.325(5.5%↓) |  0.340(4.3%↓) |
> | RoBERTa    | 0.248(27.7%↓) | 0.286(14.4%↓) | 0.203(34.1%↓) | 0.224(31.3%↓) |
>
> Table 3 above presents the outcomes of an experiment wherein we substituted various pre-trained language models as the backbone. The negligible variance in the results shows that our method is effective across different PLMs, thereby advocating that the increases of performances are not attributable to the PLMs.
>
>
> |                 |     AMI-Pk    |     AMI-Wd     |    ICSI-Pk    |    ICSI-Wd   |
> |-----------------|:-------------:|:--------------:|:-------------:|:------------:|
> | TextTiling-BERT      |     0.352     |      0.352     |     0.324     |     0.343    |
> | TextTiling | 0.412 (17.0%↓) | 0.426 (21.0%↓) | 0.426 (31.5%↓) | 0.600 (74.9%↓) |
>
> However, as the table above demonstrated, compared to our strongest baseline results (cf. [1]), we believe that the improvements brought about by each component of our method are significant.
>
> [1] Solbiati et al. 2021. Unsupervised Topic Segmentation of Meetings with BERT Embeddings.
> ***
>
> > Topic segmentation for unstructured conversations
>
> We thank the reviewer for pointing out this issue, and we have incorporated this suggestion throughout our experiment. According to the paper [2], it is more likely that the dataset we used consists of *Unstructured Chats*, transcriptions of spoken language from a spoken scene, which result in often incoherent and incomplete utterances with some word errors. In addition, we have conducted experiments on the Topical Chat, BOLT, and Wiki-727K datasets as proposed in [2], and the results are as follows:
> | Method  | Topical Chat |    BOLT   | Wiki-727K |
> |----------------------------|:------------:|:---------:|:---------:|
> | Cross Segment BERT|     0.764    |   0.569   |   0.604   |
> | Cross Segment RoBERTa-Base|     0.767    |   0.561   |   0.599   |
> | Hierarchical Bi-LSTM|   **0.951**  |   0.443   |   0.570   |
> | M3Seg|     0.950    | **0.602** | **0.612** |
>
> It is worth noting that our method is unsupervised yet still achieves equivalent or better performances than these supervised methods that use domain data for fine-tuning. Additionally, we will cite the paper [2] and add detailed analysis of topic segmentation on structured, semi-structured and unstructured conversations.
>
>
> [2] Ghosh et al. 2022. Topic Segmentation in the Wild: Towards Segmentation of Semi-structured & Unstructured Chats.
> ***
>
> > Typos and Writing Suggestions
>
> We appreciate your suggestions and have carefully checked for grammar errors, rephrased some expressions and applied a consistent format in our revised version.
>
> ***
>
> Thank you for your valuable feedback! Please let us know if you have any remaining concerns.

---

### Meta-Review · Area_Chair_PUEk · 2023-09-20

**Recommendation:** 5

**Metareview:**

This is a strong paper according to most reviewers. They value the effectiveness and novelty of the method, the proper referencing, clear explanations and motivations.  Several of the issues brought up by reviewers were properly addressed by the authors, for example, they conducted experiments on the datasets proposed in Gosh et al.’s work. All in all, the authors did a good job in addressing the reviewer’s concerns and building trust in being able to improve the final version of this paper, which resulted in some scores being raised. There are four reviewers for this paper as one of the three initial reviewers explained that the topic was outside their field of expertise. I am basing my meta-review on the three other reviewers. Both soundness and excitement scores are close to 4 on average.

---

### Decision · Program_Chairs · 2023-10-07

**Decision:**

Accept-Main

**Comment:**

This is a strong paper according to most reviewers. They value the effectiveness and novelty of the method, the proper referencing, clear explanations and motivations.  Several of the issues brought up by reviewers were properly addressed by the authors, for example, they conducted experiments on the datasets proposed in Gosh et al.’s work. All in all, the authors did a good job in addressing the reviewer’s concerns and building trust in being able to improve the final version of this paper, which resulted in some scores being raised. There are four reviewers for this paper as one of the three initial reviewers explained that the topic was outside their field of expertise. I am basing my meta-review on the three other reviewers. Both soundness and excitement scores are close to 4 on average.